# Study on the Shape of the Aerator of High-Head Discharge Tunnel with Mild Bottom Slope

**Xueyu Zheng** [1,2,3], **Luchen Zhang** [2], **Shiqiang Wu** [1,2,*] **and Kai Song** [2]

1 State Key Laboratory of Water Resources and Hydropower Engineering Science, Wuhan University, Wuhan 430072, China; xyzheng@whu.edu.cn
2 State Key Laboratory of Hydrology-Water Resources and Hydraulic Engineering, Nanjing Hydraulic Research Institute, Nanjing 210029, China; lczhang@nhri.cn (L.Z.); Songkaihekele@163.com (K.S.)
3 Hydropower and Hydraulic Engineering Institute, Power China Guiyang Engineering Corporation Limited, Guiyang 550081, China
\* Correspondence: sqwu@nhri.cn; Tel.: +86-138-5156-8908

**Abstract:** Due to the high flow velocity and easy cavitation of high-head drainage tunnels, it is usually necessary to set up aeration facilities. In particular, when the bottom slope of the tunnel is mild, the aeration facilities often have problems such as difficulty with air intake, short cavity, and serious water accumulation, which aggravate the risk of cavitation damage. In this paper, based on the Rumei hydropower station and the Gushui hydropower station, a method combining theoretical analysis and model testing is used to solve the connection problem between the aeration facility and the 3% mild bottom slope of a tunnel body, and the aeration facility shape of "lifting ridge + flat (mild) slope + steep slope" is put forward. The research shows that the steep slope section can smoothly connect the water flow over the cantilever, reduce the jet impact angle, prevent the water from backtracking, and produce a long and stable cavity in the flat (mild) slope section. The aeration concentration along the bottom of the tunnel is higher than 3% at 140 m over the top of the dam. The aeration effect of this type is better, and it can provide effective long-distance protection for a drainage tunnel with high head and a mild bottom slope.

**Keywords:** mild bottom slope; aeration facility; water flow connection; cavity length; aeration concentration





## 1. Introduction

Cavitation erosion was first found in military destroyers and turbine propellers and appeared 30 years later in high-head drainage structures of hydropower projects. Cavitation erosion damage is the main hydraulic safety risk faced by high-head drainage tunnels, which seriously endangers the safe operation of drainage buildings. Aeration erosion reduction is one of the important measures used to reduce the cavitation erosion damage of high-speed flow. Many experts have specifically described the problems of aeration corrosion reduction technology and collected a large number of examples [1,2]. There are countless special studies on the aeration problem of high-head spillways. Wu et al. [3] and Xu et al. [4], respectively, studied the necessity of setting aeration facilities on the clear section of a spillway and the surface of a step spillway for specific projects. Dong et al. [5,6] studied the variation law of pressure and cavitation number with air concentration, pressure waveform, and degree of cavitation erosion under both aerated and unaerated conditions. Meng et al. [7] concluded through experiments that adding a side deflector at the outlet of an aerator can increase the cavity length of the aerator, which is affected by the aerator's vertical plunge and lateral expansion. H. Chanson [8] analyzed the relationship between air flow and pressure in the cavity under nappe. M. Cihan Aydin [9] studied the air entrainment rate and air concentration distribution law in aeration facilities. Muhammad Kaleem Sarwar [10] found that different sizes of vents and different oblique angles of aeration facilities have significant influences on dimensionless performance indicators of

aeration facilities. Cui et al. [11] studied the aeration performance of an inverted umbrella aerator and bubble characteristics in the aeration tank under different conditions, revealing the internal relationship between bubble characteristics and aeration performance. Juan César Luna-Bahena et al. [12] studied the combined effect of the top pier and bottom aeration facilities of a smooth spillway. Bai et al. [13–15] studied the characteristics of microscopic bubbles near the bottom of a drain tank, revealed the flow characteristics of air and water at the bottom, and proposed a self-similar relationship between air concentration and bubble frequency in various regions. Bai et al. [16] proposed a method to estimate the air entrainment rate of a slot bed by considering the effective air cavity length and the influence of air escaping the jet. Through research, Kristian Kramer et al. [17] proved that the minimum air concentration is a function of the Froude number, and the minimum air concentration is restricted by the starting point of air. Michael Pfister et al. [18,19] pointed out that there are three flow zones in aerated water flow: (1) the jet zone, (2) the reattachment spray zone, and (3) the far-field zone. Aeration facilities mainly affect the average air concentration, and the main parameters affecting the downstream air transport of the aeration facilities are the near-flow Froude number, the deflection angle, and the bottom angle of the vent.

Although there are many research results, most have been obtained on spillway tunnels with a steep bottom slope. However, for a spillway tunnel with a mild bottom slope, when the water head is high, the Fr number is low, and the aeration facility is prone to problems such as cavity instability, serious water accumulation, and difficulty with air intake, it cannot effectively perform its aeration erosion reduction function. In certain settings, it is also easy to form a new source of cavitation. The selection of aerator type is not only affected by the flow velocity, the single-width flow velocity, and the bottom slope of the flow surface, but also by the flow conditions. It is difficult to adapt the same aerator type to many projects, and the research results are poor in general. Some Chinese scholars carried out research on the aerator types of discharge tunnels with a mild bottom slope based on specific projects. Through model tests, Zhou et al. [20] proposed a kind of aerator arrangement with airfoil lifting, which effectively solved the problems of short cavity and backwater in the cavity. Liu et al. [21] found that a three-dimensional convex aeration hump can better reduce the backwater phenomenon of the cavity and achieve the effect of aeration. Wang [22] concluded that the aeration effect of sudden expansion and sudden fall of aeration hump is mainly affected by two "water accumulation" sources and one "water retreat mechanism", and the impact angle of the jet played a decisive role in the backflow of water and water retreat. An aeration method and an optimized shape of small bottom slope with local slope change were proposed. Pang et al. [23] studied and proposed a novel U-shaped trough aerated cantilever, which ensured the cavity stability of an open-flow drainage tunnel with a small bottom slope. Wu and Wang et al. [24] proposed a combined aeration form of "swallow tail lifting ridge + stick slope" and found that it could effectively solve the problem of water accumulation in an aerated cavity with a small bottom slope and low Fr water flow. In addition, some scholars have studied the hydraulic characteristics of stepped spillways with a pre-aerated ramp and found that the ability of dam surface aeration erosion reduction can be improved by adjusting the parameters of the ramp type [25,26]. Scholars have put forward a U-shaped groove aeration ceiling [23], a V-shaped aeration ceiling [27], an airfoil tip ceiling [20], a plane concave–convex aeration ceiling [28], and a three-dimensional aerator with uniform horizontal and longitudinal changes [29], etc. The types of aerators above have significantly improved the cavity length, aeration effect, and protection length of aeration facilities with a bottom slope of 5–10%. However, when the bottom slope of a tunnel is lower than 5%, it is more difficult to connect the aeration facilities with the mild bottom slope of the downstream non-pressure tunnel. The type of connection is essential to determine whether the aeration facilities can play a role or not.

Relying on the bottom slope of about 3% in the Rumei hydropower station and the Gushui drainage tunnel of hydropower station, through the method of combining

theoretical analysis and model testing, studies of the connection pattern between the aeration sill and the downstream bottom slope were carried out. We propose an aeration facility shape of "lifting ridge + flat (mild) slope + steep slope", which is suitable for a high-head drainage tunnel with a mild bottom slope, ensures the stability of the aeration cavity, and improves the adaptive conditions.

## 2. Materials and Methods

The spillway tunnel of the Rumei hydropower station adopts a short pressurized inlet. The floor elevation of the intake is 2827.00 m, and the orifice size of the working gate is 7 m × 13 m. The total length of the horizontal projection of the axis is 1010.58 m, of which the length of the non-pressure tunnel section is 714.66 m, the bottom slope is i = 3.0%, the section type of the tunnel body is the city gate type, and the section size is 11.00 m × 15.17 m. The outlet of the non-pressure tunnel is connected with the grip curve, the steep groove section, and the energy dissipation section. The spillway tunnel has a maximum operating head of 69.42 m, a maximum discharge capacity of 2816 m³/s, a maximum flow velocity of 32 m/s, and an Fr number of less than 4. It is a typical spillway tunnel with a high head and a mild bottom slope. The aeration facility of the original cave body is a drop type, with a drop height of 2.0 m and a vent size of 1.5 × 2.0 m, as shown in Figure 1a.The hydraulic characteristics of the aeration facilities were observed by model testing. The model test was designed using the gravity similarity criterion, and the geometric scale Lr = 80. Parameters related to model test list in Table 1.

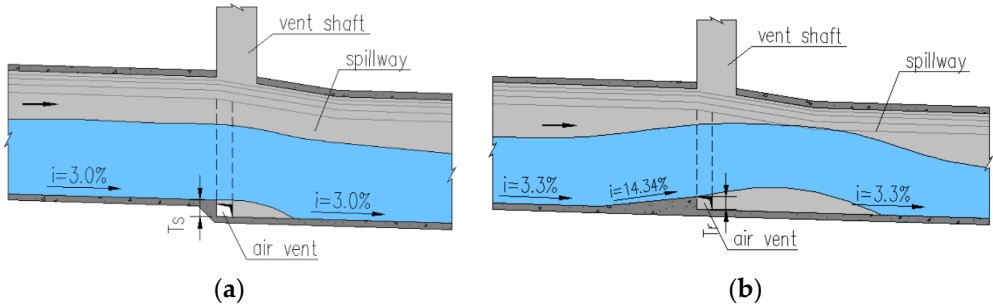

**Figure 1.** The original aeration type: (**a**) Rumei spillway tunnel; (**b**) Gushui empty sand flushing tunnel.

**Table 1.** Parameters related to model test.

| Name of Power Station | Model Test Object | Total Length (m) | Length of Non-Pressure Section (m) | Bottom Slope of Non-Pressure Section i (%) | Maximum Operating Head (m) | Maximum Flow (m³/s) | Maximum Velocity (m) | Model Scale Lr |
|---|---|---|---|---|---|---|---|---|
| Rumei | spillway tunnel | 1010.58 | 714.66 | 3 | 69.42 | 2816 | 32 | 1:80 |
| Gushui | empty sand flushing tunnel | 2508 | 1889.5 | 3.3 | 105 | 1269 | 32.8 | 1:50 |

The empty sand flushing tunnel of the Gushui hydropower station is composed of an inlet section, access gate chamber, pressurized section, underground working gate chamber, non-pressurized section, and flip lifting ridge section. The total horizontal length is about 2508 m, of which the pressurized tunnel section accounts for 468 m. The bottom slope is a flat bottom slope with an inner diameter of 9.5 m. The length of the non-pressurized tunnel is 1889.5 m, the bottom slope is 3.3%, and the section is a square and circular shape of 9.5 × 16 m. The intake floor elevation is 2162 m, with a flared inlet. There is an 8 × 10 m flat plate accident access door in the well. A working gate chamber is arranged at the end of the pressurized cavity section, which is equipped with a 7.5 × 8.0 m arc working door and is followed by a pressurized cavity section. An aerator and aerator well are set every 150 m in the pressurized cavity section. The outlet end adopts a flip bucket for energy dissipation, and the elevation of the flip bucket is 2096.143 m. The maximum running

water head is 105 m, the maximum discharge capacity is 1269 m$^3$/s, the maximum flow velocity is 32.8 m/s, and the Fr number is less than 4. It is a typical drainage tunnel with a high water head and a mild bottom slope. The aeration facilities of the original cave body are of the bucket type, with the height being Tr = 1.0 m and the size of the vent being 1.5 × 1.5 m, as shown in Figure 1b. The hydraulic characteristics of the aerator were observed by model testing. The model test was designed using the gravity similarity criterion, and the geometric scale Lr = 50. Parameters related to model test list in Table 1.

## 3. Results and Discussion

### 3.1. Analysis of Flow Characteristics of Aeration Facilities

When the water level and gate are fully open in the original design, the aeration facility of the Rumei spillway tunnel is only set with falling, and the aeration cavity is completely filled with water, thus blocking the air inlet channel, and the aeration function of the aeration facility is ineffective (Figure 2a). Under the condition of normal storage level of the original body and full opening of the gate, the aerator of the Gushui empty sand flushing tunnel is only set up with lifting, which cannot form a stable cavity (Figure 2b).

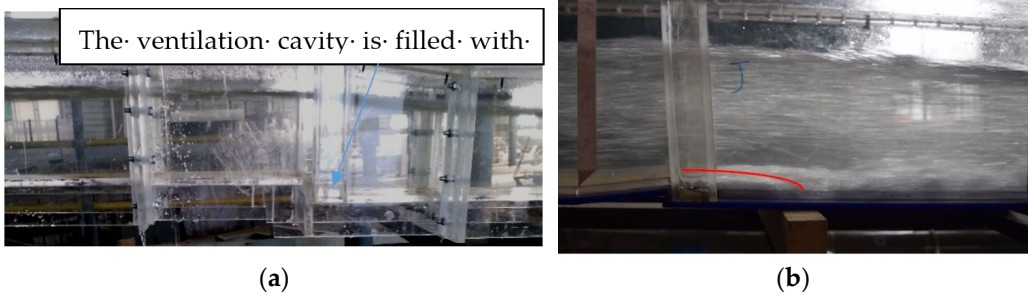

(a)   (b)

**Figure 2.** Flow pattern of aerator: (**a**) Rumei spillway tunnel; (**b**) Gushui empty sand flushing tunnel.

According to the above test results, the analysis shows that the body of the spillway tunnel with a high head and a mild bottom slope adopts aeration facilities, and the flow over the dam has the following characteristics.

(1) The effect of gravity is obvious, and the jet impact angle is large: the slope of the spillway tunnel on the mild bottom slope is below 10°, and the Fr number is generally lower than seven. Under the flow condition of low Fr number, the effect of aeration resistance can be ignored, but the influence of gravity is very significant, the water is more dispersed, and the water easily enters the aerated cavity. Before flow picking, due to the restriction of the bottom plate, the influence of gravity on the convection state is very small. After the water passes through the ridge, the water moves in a parabolic motion under the action of gravity. The curvature of the lower edge surface of the jet tongue is large, and the impact angle between the jet and the mild bottom slope at the drop point is large (generally greater than the 'critical value'). The formation of the backwater in the cavity after the aerator is closely related to the impact angle of the jet flow at the end of the cavity [30], leading to the backtracking of the water flow and the formation of water accumulation.

(2) The space layout of the aerator is limited: the bottom slope of the spillway tunnel is small, which leads to the limited space arrangement of the aeration facilities. In order to ensure a certain cavity, the plunging type needs to have a higher drop difference. However, in this case, the horizontal distance between the bottom slope and the original tunnel is far, and the separated water more easily falls in the connecting section, which makes the cavity unstable and causes more backflow.

To sum up, the key considerations for the arrangement of aeration facilities in a spillway tunnel with a mild bottom slope and a high head are as follows: (1) the choice of lifting or falling under the space layout limitation and (2) whether the size of the jet impact angle can prevent the flow from backtracking and form a stable cavity. Therefore, the appropriate lifting and dropping type and jet impact angle are the key to the arrangement

of aeration facilities in the mild bottom slope spillway tunnel, which puts forward higher requirements for the lifting form of aeration and downstream connection mode.

### 3.2. Influence of the Type and Height of the Aerator on the Characteristic Parameters of Water Flow

In view of the problems existing in the original aeration facilities of the spillway tunnel of the Rumei hydropower station and the empty sand flushing tunnel of the Gushui hydropower station, the structure of the original aerator was optimized. For a mild bottom slope discharge tunnel, setting the carry camp alone or drop of the aerator cavity are serious backwater problems. Therefore, without considering other factors, it is necessary to first confirm whether the lift or the fall is more likely to produce a larger cavity, and then make a theoretical analysis of the influence of the lift height on the cavity length (Figure 1).

The falling heights $T_s$ were set at 0, 1.0, 1.5, and 2.0 m, and the lifting heights $T_r$ were set at 0, 1.0, 1.5, and 2.0 m. The parameters of the different lifting and falling shapes and the working conditions are listed in Table 2. By comparing the cavity length calculation formulas of Wu [31] and Luo [32], Luo's cavity length calculation formula and Yang's jet impact angle calculation formula [33] were adopted to study the cavity length and jet impact angle of different kinds of sills. According to Luo [32], Equation (1) is based on the analysis of a micro element in the flow zone at the bottom edge of the jet, and according to the characteristics of the turbulent free jet, ignoring the secondary forces, the main forces on the micro element are gravity, resistance, pressure difference force, and inertia force. The motion equation of the microelement is established, and the calculation formula of the cavity length $L_{jet}$ is obtained after necessary simplification. $L_{jet}$ represents the distance between the end of the aerator and the falling point of the flow. The formulae are as follows:

$$L_{jet} = V_1 \cos(\varphi_1)T + 0.5g\left(\sin \alpha - 0.00625 F_r^2\right)T^2 \tag{1}$$

$$T = \frac{V_1 \sin \varphi_1}{g(\cos \alpha + P_N)}\left[1 + \sqrt{1 + \frac{2(T_r + T_s)g(P_N + \cos \alpha)}{(V_1 \sin \varphi_1)^2}}\right] \tag{2}$$

where

$L_{jet}$ is the simplified cavity length (m);
$\varphi_1$ is the actual efflux exit angle, and its value will be less than the cantilever slope, $\varphi$; $\varphi_1 = \xi_1 \varphi$;

$$\xi_1 = \sqrt{\frac{e^x - e^{-x}}{e^x + e^{-x}}} \tag{3}$$

$\alpha$ is the bottom slope of a groove;
$V_1$ is the actual flow velocity at the bottom edge of the jet (m/s), $V_1 = \xi_2 V$, and the velocity correction coefficient is 0.96;
$P_N$ is the cavity negative pressure index, $P_N = \frac{\Delta P_C}{\rho_W g h}$, where $\Delta P_C = P_a - P_c$, the cavity negative pressure is $P_C$, the atmospheric pressure is $P_a$, $\rho_W$ is the fluid density (kg/m$^3$), and $g$ is the gravitational acceleration, (m$^2$/s).

$$\theta_{min} = \tan^{-1}\left(V_y / V_x\right) \tag{4}$$

$$V_x = V_1 \cos \varphi_1 + \left(\sin \alpha - \beta F_r^2\right)gT \tag{5}$$

$$V_y = V_1 \sin \varphi_1 - (\cos \alpha - P_N)gT \tag{6}$$

where

$\theta_{min}$ is the jet angle of impact (°);
$\beta$ is the drag coefficient, which is 0.00625.

The cavity lengths calculated with different lifting and falling shapes are shown in Figure 3. The cavity length increases with the increase in the ridge height. When the heights are the same, the cavity length corresponding to the cantilever coping sill is larger

than that of the falling sill. When the relative slope is 10%, the cavity length corresponding to the cantilever coping sill is generally about 12 m longer than that of the falling sill. With the increase in the ceiling height, the increase range of the cavity length corresponding to the cantilever coping sill is larger than that of the falling sill. Group 1 is the original body type, and the calculation of the cavity length is short. Due to the small slope of the cavity body, the intersection angle between the flow over the ridge and the bottom plate is large, and effective aeration cannot be obtained. Therefore, when the layout of the spillway tunnel on the mild bottom slope is limited, the cantilever should be given priority as the basic shape of the aeration facility in order to form a larger aerated cavity.

**Table 2.** Theoretical calculation and analysis of body type group table.

| Group | Average Velocity of Cross Section on the Ridge, V (m/s) | Slope of Trough Bottom, $\alpha$ | The Fr Number | Height of Lifting Ridge, $T_r$ (m) | Drop Height, $T_S$ (m) | Water Depth, h (m) | Cavity Length, Ljet (m) | Jet Angle of Impact (°) |
|---|---|---|---|---|---|---|---|---|
| 1 | 31.900 | 0.030 | 3.621 | 0.000 | 2.000 | 7.910 | 19.455 | 11.68 |
| 2 | 31.900 | 0.030 | 3.621 | 0.000 | 1.500 | 7.910 | 16.861 | 10.13 |
| 3 | 31.900 | 0.030 | 3.621 | 0.000 | 1.000 | 7.910 | 13.779 | 8.29 |
| 4 | 31.900 | 0.030 | 3.621 | 0.000 | 0.5 | 7.910 | 9.754 | 5.87 |
| 5 | 31.900 | 0.030 | 3.621 | 0.000 | 0.000 | 7.910 | 0.000 | 0.00 |
| 6 | 31.900 | 0.030 | 3.621 | 0.500 | 0.000 | 7.910 | 22.504 | 18.81 |
| 7 | 31.900 | 0.030 | 3.621 | 1.000 | 0.000 | 7.910 | 25.977 | 20.89 |
| 8 | 31.900 | 0.030 | 3.621 | 1.500 | 0.000 | 7.910 | 28.574 | 22.33 |
| 9 | 31.900 | 0.030 | 3.621 | 2.000 | 0.000 | 7.910 | 30.848 | 23.56 |

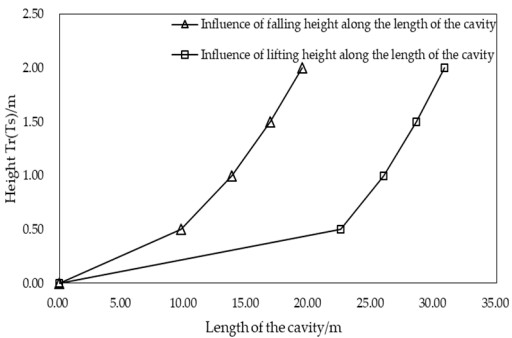

**Figure 3.** Relationship between different lifting and cavity length.

As can be seen from Figure 4 and Table 2, the jet impact angle increases with the increase in the ridge height. For the same height, the impact angle of the jet corresponding to the cantilever coping sill is larger than that of the falling sill. With the increase in the height, the increase range of the jet impact angle corresponding to the cantilever coping sill is larger than that of the falling sill. When the relative slope is 10%, the impact angle of the jet corresponding to the Camby bump is approximately 12–13°. The impact angle of the lifting jet increases by approximately 1.2–2.0° and the impact angle of the falling jet increases by approximately 1.6–2.4° with the increase in the ridge height by 0.5 m. With the increase in height, the increasing range of the jet impact angle becomes smaller. According to the study of Zhang et al. [30], the depth of water in the cavity shows an overall trend of increase with the increase in jet impact angle. From the analysis of the jet impact angle, it can be seen that special attention should be paid to the connection between the aeration dam and the downstream when selecting the aerator shape so as to reduce the jet impact angle as much as possible and prevent backwater in the cavity.

On the basis of determining the advantages of flip bucket in aerating facilities of flood discharge tunnel with mild bottom slope, we also analyzed the influence of flip bucket slope on cavity length. On the basis of group 7 above, we set different cantilever slopes of 5%, 10%, 15%, 20%, 25%, and 30%, respectively, to study the variation law of cavity length.

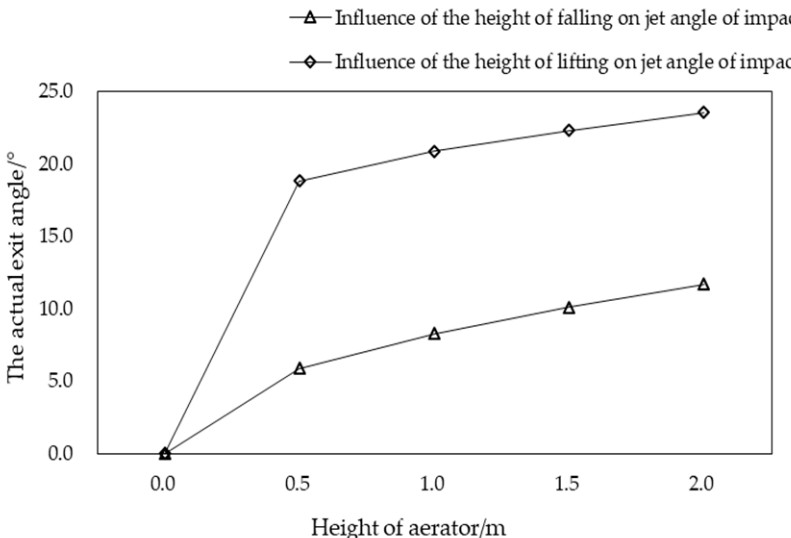

**Figure 4.** Relationship between different tip and jet impact angle.

It can be seen from Figure 5 that the cavity length has a good linear relationship with the cantilever slope under the same other conditions. The cavity length increases with the increase of the cantilever slope, and the cavity length increases by 7 m for every 5% increase of the flip bucket slope. Similarly, there is a good linear relationship between the cantilever slope and the jet impact angle, and the cavity length increases by 6° with the increase of the cantilever slope by 5%.

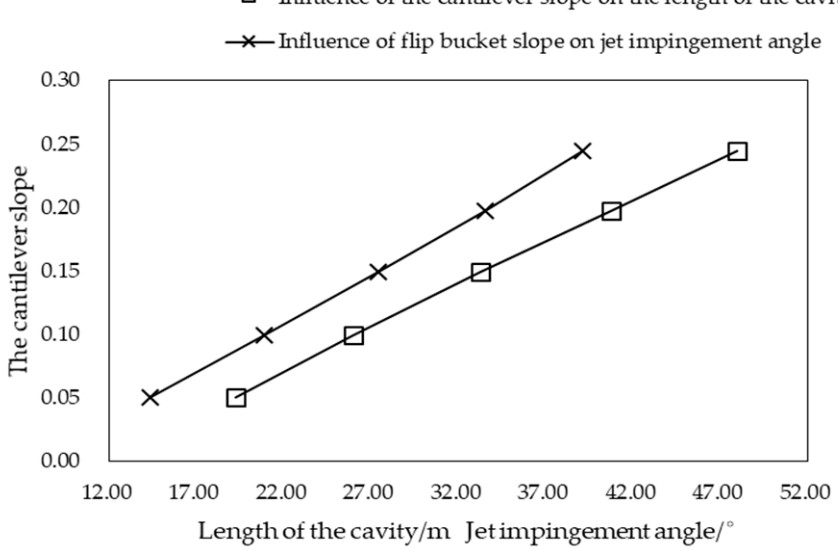

**Figure 5.** Relationship between different the cantilever slope and cavity length.

### 3.3. Mild Bottom Slope Aeration Facility Shape

The experiment proved that whether the original scheme of the spillway tunnel of the Rumei hydropower station and the empty sand flushing tunnel of the Gushui hydropower station adopt either the lift or the fall shape, the aerated cavity is completely filled with backwater. Combined with the theory analysis, a mild slope discharge tunnel should first use the lifting bucket type; at the same time, the downstream slope cohesive form should be set up and the jet point should be controlled to keep the jet impact angle within 10°. Therefore, the aerator shape suitable for a mild bottom slope is shown in Figure 6. According to its shape characteristics, it is described as a shape of "lifting ridge + flat (mild)

slope + steep slope". Among them, the role of the lifting ridge is to project water a certain way, which is conducive to the formation of a larger cavity range; the role of the steep slope is to smooth the connection of the projected water, reduce the jet impact angle, and reduce the backflow water; and the flat (mild) slope is the place where the stable cavity is formed.

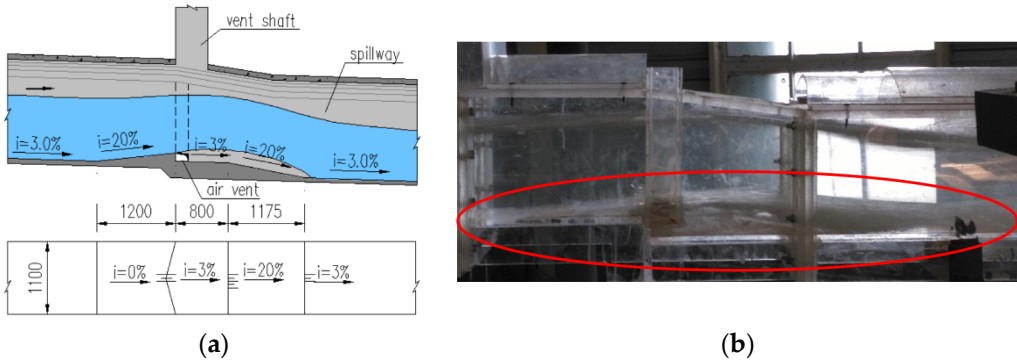

| (a) | (b) |

**Figure 6.** Shape design and flow pattern (Unit: cm): (**a**) the aerator shape suitable for a mild bottom slope; (**b**) flow pattern of gentle bottom slope.

### 3.4. Validation of Aeration Effect

#### 3.4.1. Flow Pattern

For the spillway tunnel of the Rumei hydropower station, the initially proposed values were 10% lifting ridge + 3% mild slope + 20% steep slope. This paper verifies and analyzes the connection effect of the aerator facilities. For the 10% lifting ridge + 1.2 m high ridge + 8 m long 3% mild slope + 20% steep slope type, due to the significant influence of gravity and side wall resistance in the case of a small bottom slope, there are differences in the parabolic curvature and tip distance along the center and both sides of the lower edge surface of the projectile water after conventional 2D straight line lifting. The center of the projectile distance is far and high, the two sides are close and low, the drop point has convex distribution, it is easy to produce a rotary water flow, and the end of the cantilever adopts the dovetail structure. The design scheme and test results are shown in Figure 6. After the water flows out of the aerator, the water surface is stable, a certain length of cavity is formed at the aerator, and the water surface line does not reach the circular segment of the city gate. However, due to the high velocity in the spillway tunnel and the long distance between the water tongue and the drop point at the end of the steep slope, or even the bottom floor of the cavity, the jet impact angle is as large as 20.5°. As a result, there is still a serious amount of water in the cavity.

Based on Group 10, the horizontal length of the mild slope section in Group 11 was extended to 16 m, and the other parameters remained unchanged. The design scheme and test results are shown in Figure 7. After optimization, the drop point of the water tongue was located in the steep slope section, and the impact angle of the jet was 5°. The aerator could form a stable cavity without water accumulation, and the length of the cavity was about 23.2 m. The flow pattern connection of the aeration facility was good.

For the empty sand flushing tunnel of the Gushui hydropower station, a total of 12 aerators with the shape of "lifting ridge + flat (mild) slope + steep slope" were set. Under normal water level conditions, the water flow after the aeration hump was smooth, the steep slope section was well connected, and the flat slope section formed an effective and stable cavity, which completely eliminated the water accumulation in the cavity, and the length of the cavity was 15 m. The shape of aeration facility no. 1 adopted 14.5% spike + 12 m long flat slope +16% steep slope. The flow patterns are shown in Figure 8. The shapes of aerators No. 2–12 were the same, with slight differences in size (Table 3). Some of the flow patterns are shown in Figure 9.

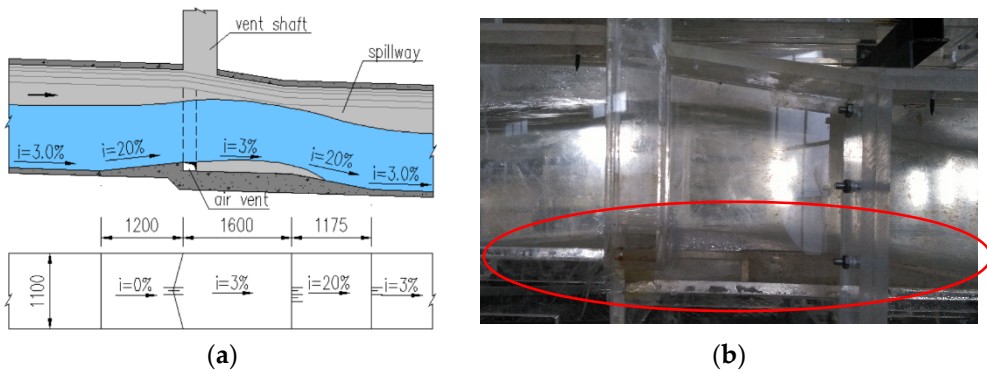

**Figure 7.** Group 11 shape design and flow pattern (Unit: cm): (**a**) the design aerator shape of Rumei; (**b**) flow pattern.

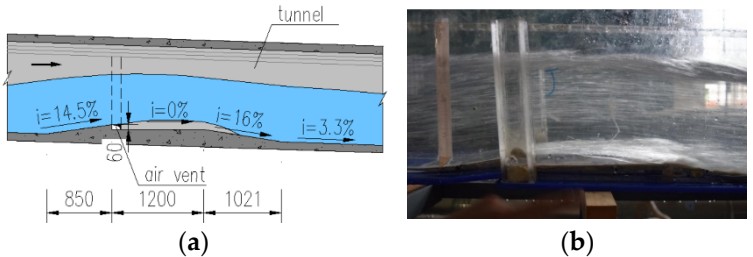

**Figure 8.** Optimal shape and flow pattern of aerator No. 1 (Unit: cm): (**a**) the design aerator shape of Gushui; (**b**) flow pattern.

**Table 3.** Shape parameters of aerators No.1–12.

| No. | Sill with Cantilever Coping | | | Flat (Mild) Slope | | Steep Slope | |
| --- | --- | --- | --- | --- | --- | --- | --- |
| | Horizontal Length (m) | Slope (%) | Lifting Ridge Height (m) | Horizontal Length (m) | Slope (%) | Horizontal Length (m) | Slope (%) |
| 1 | 8.5 | 14.5 | 0.6 | 12 | 0 | 10.21 | 16 |
| 2 | 8.5 | 14.5 | 0.7 | 14.5 | 0 | 10.08 | 16 |
| 3 | 8.5 | 14.5 | 0.7 | 14.5 | 0 | 10.08 | 16 |
| 4 | 8.5 | 14.5 | 0.7 | 14 | 0 | 9.95 | 16 |
| 5 | 8.5 | 14.5 | 0.7 | 13.5 | 0 | 9.82 | 16 |
| 6 | 8.5 | 14.5 | 0.7 | 13.5 | 0 | 9.82 | 16 |
| 7 | 8.5 | 14.5 | 0.7 | 13 | 0 | 9.69 | 16 |
| 8 | 8.5 | 14.5 | 0.7 | 12.5 | 0 | 9.56 | 16 |
| 9 | 8.5 | 14.5 | 0.7 | 12.5 | 0 | 9.56 | 16 |
| 10 | 8.5 | 14.5 | 0.7 | 12 | 0 | 9.43 | 16 |
| 11 | 8.5 | 14.5 | 0.7 | 10 | 0 | 8.91 | 16 |
| 12 | 8.5 | 14.5 | 0.7 | 10 | 0 | 8.91 | 16 |

### 3.4.2. Aeration Concentration

The biggest problems of aeration facilities in a mild slope spillway tunnel are backwater in the cavity and inefficient aeration [34]. Therefore, a wide cavity can be generated according to the above body type. Next, it is necessary to verify whether the aeration concentration along the passage meets the requirements. The scale of the spillway tunnel model of the Rumei hydropower station is 1:80, so it is difficult to accurately reflect the aeration effect of the aerator. Therefore, a 1:50 Gushui hydropower station sand-washing tunnel model was used to verify the applicability of the aerator, and the aeration concentration distribution along the tunnel behind the aerator under a normal storage level was measured.

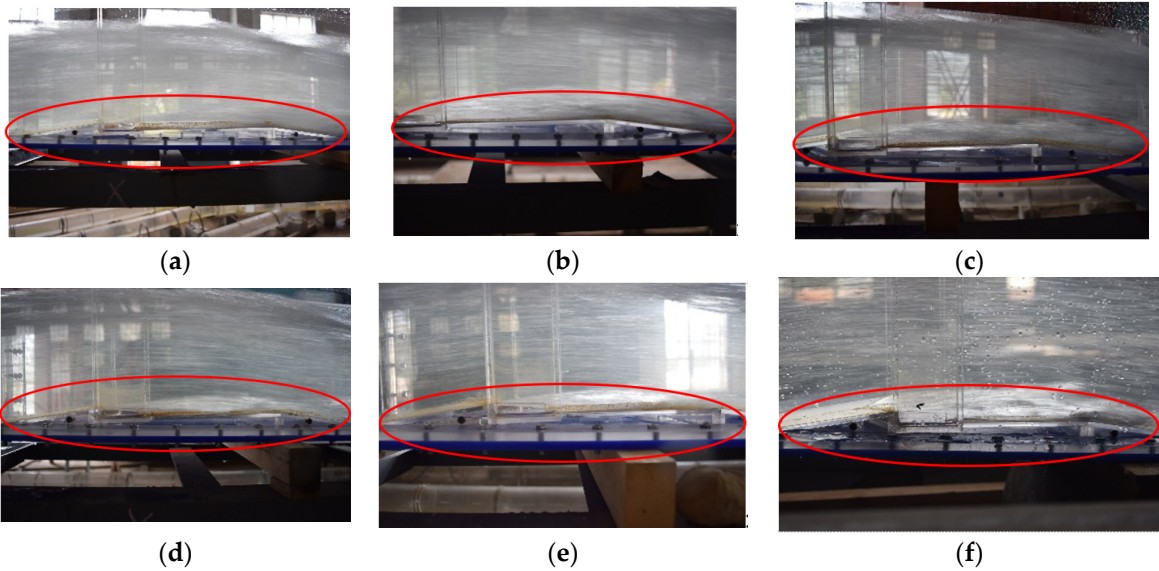

**Figure 9.** Flow pattern of parts of aerators 1–12: (**a**) aerator 2; (**b**) aerator 4; (**c**) aerator 6; (**d**) aerator 8; (**e**) aerator 10; (**f**) aerator 12.

Figure 10a shows the bottom-side aeration concentration distribution at the bottom of aerator 1. Figure 10b shows the bottom-side aeration concentration distribution in aerators 2~12. Studies show that [35] the air content of the water flow at the end of the cavity is partly caused by the mixing of the lower edge of the water column. The other one-third is swept in when the water plumes hit the floor. As shown in Figure 10, the aeration amount within 15 m before the aerator is due to the turbulent mixing of the water–air interface in the process of propelling the water tongue. Due to the existence of stable cavities, both the upper and lower surfaces of the water tongue are mixed with air, resulting in a large amount of aeration. The extrusion between the main stream and the bottom plate at the drop point of the water tongue hinders water intake and leads to a decrease in aeration concentration near the bottom before and after the drop point of the water tongue. In the optimization scheme, the aeration concentration along the tunnel is consistently greater than 3% [36] within 140 m after the aerator. The total length of the non-pressurized section is 1889.5 m, and 12 aerators are enough to provide full aeration protection, so that the aeration concentration in the non-pressurized section is greater than 3%, and the risk of cavitation and cavitation in the non-pressurized section is greatly reduced. The aeration concentration of the original scheme starts to fall below 3% after the aeration ridge of 40 m, indicating that the aeration facility of the "lifting ridge + flat (mild) slope + steep slope" shape has a better aeration effect and can provide long-distance aeration protection.

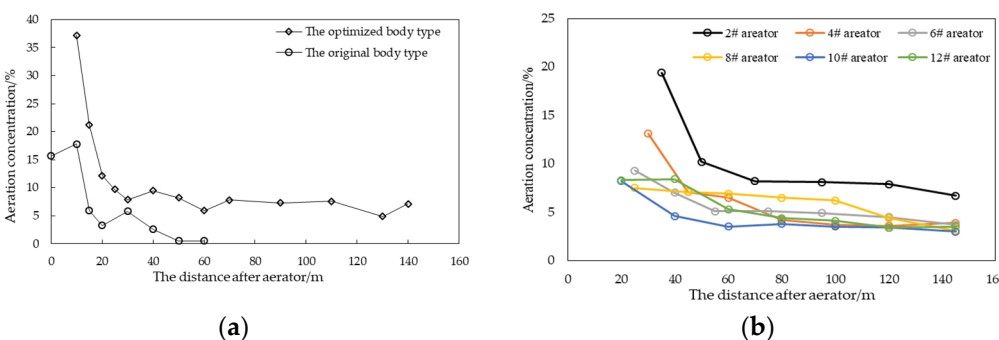

**Figure 10.** The aeration concentration distribution after the aerator: (**a**) aerator 1; (**b**) aerators 2–12.

## 4. Conclusions

(1) The cavity length of the aerated body increases with the increase in the lifting height; at the same height, the cavity length corresponding to the cantilever coping sill is larger than that of the falling sill. When the relative slope is 10%, the cantilever coping sill is generally about 12 m longer than the falling sill. With the increase in the height, the increase in the cavity length corresponding to the ceiling is more significant than that of the ceiling. When the space of aeration facilities in the mild bottom slope discharge tunnel is limited, it is advisable to give priority to the lifting type as the basic shape of aerator in order to form a wider aeration cavity.

(2) The recommended shape of aerator for a mild bottom slope discharge tunnel should be the type of "lifting ridge + flat (mild) slope + steep slope". This type of lifting ridge causes the water to project to a certain extent, forming a wide cavity. The flat (mild) slope is the recommended position to form a stable cavity. The steep slope can achieve smooth connection of the ejected water, reduce the impact angle of the jet, and reduce the backflow of water in the aerated cavity.

(3) The aeration facilities with the "lifting ridge + flat (mild) slope + steep slope" shape have a stable aeration cavity, and the aerator effect along the tunnel is good. The aeration concentration along the tunnel is consistently greater than 3% within 140 m after the aerator. The layout can enable an aeration concentration all along the spillway tunnel of greater than 3%, which meets the specification requirements and has good popularization and application value.

**Author Contributions:** Conceptualization and methodology, S.W. and L.Z.; validation and test, X.Z. and K.S.; writing—original draft preparation, X.Z.; writing—review and editing, K.S.; supervision and funding acquisition, S.W. All authors have read and agreed to the published version of the manuscript.

**Funding:** This research was mainly financed by the Central Public Welfare Research Institute (Y120001, Y121011), science and technology projects of China Power Construction Corporation Limited (DJ-ZDXM-2017-05), and Guizhou Science and Technology Support Plan: Guizhou Science and Technology Joint Support (2019) No. 2890.

**Institutional Review Board Statement:** Not applicable.

**Informed Consent Statement:** Not applicable.

**Data Availability Statement:** Not applicable.

**Acknowledgments:** The authors sincerely thank the reviewers who contributed their expertise and time for reviewing this manuscript.

**Conflicts of Interest:** The authors declare no conflict of interest.

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
