# Peer review of "Study on the Shape of the Aerator of High-Head Discharge Tunnel with Mild Bottom Slope"

_water, doi:10.3390/w13152128_

Round 1

Reviewer 1 Report

This paper studied the shape of the aerator of high-head discharge tunnel with mild bottom slope, using a method of combining theoretical analysis and experimental model testing. The proper type of aeration effect was found for the RM hydropower station and the GS hydropower station. This research result would be helpful for the design of hydropower station. Though, there are some problems need to be improved before publication.
1.What is the full name of RM and GS hydropower stations?
2.At line 96 of the introduction, this sentence has grammar error: "The type of connection is key to whether the aeration facilities can play a role or not. " Better to add it as "The type of connection is essential to determine whether the aeration facilities can play a role or not." Please carefully check your manuscript and improve your writing.
3.Figure 1 and Figure 2 look similar to Figure 4. Why bother to show them again? You only need to sign those important parameter and then explain their difference.
4.There are two figures of Figure 4. Please correct it.
5.In the second Figure 4, what does it mean of Height Tr(Ts)? Do you mean that Tr is a function of Ts?
6.How to use Eq.(1) to Eq.(6) to calculate out your resulst in Table 1. Also, I do not quite understand what does 'Primary form and comparative optimization of the form group schedule' mean.
7.In Table 1, is the Jet angle of impact same as the impact angle between the jet and bottom plate in Eq.(4)?
8.Your writing skills really need to be improved!

Reviewer 2 Report

please see attached

Round 2

Reviewer 2 Report

The reviewer's comments have been responded with efforts, the overall quality of the manuscript has improved although the quality of the figures still has  plenty space to improve, but I think this should be considered by the authors in their future work particularly on how to better presenting the flow patterns.  

Please do further proof-reading as there have been quite substantial changes in the revised manuscript.